# Fusion Tag Design Influences Soluble Recombinant Protein Production in *Escherichia coli*

**DOI:** 10.3390/ijms23147678

**Published:** 2022-07-12

**Authors:** Christoph Köppl, Nico Lingg, Andreas Fischer, Christina Kröß, Julian Loibl, Wolfgang Buchinger, Rainer Schneider, Alois Jungbauer, Gerald Striedner, Monika Cserjan-Puschmann

**Affiliations:** 1Austrian Centre of Industrial Biotechnology, Muthgasse 18, 1190 Vienna, Austria; christoph.koeppl@students.boku.ac.at (C.K.); nico.lingg@boku.ac.at (N.L.); andreas.fischer@boku.ac.at (A.F.); christina.kroess@uibk.ac.at (C.K.); julian.loibl@boku.ac.at (J.L.); rainer.schneider@uibk.ac.at (R.S.); alois.jungbauer@boku.ac.at (A.J.); gerald.striedner@boku.ac.at (G.S.); 2Department of Biotechnology, Institute of Bioprocess Science and Engineering, University of Natural Resources and Life Sciences, Muthgasse 18, 1190 Vienna, Austria; 3Center for Molecular Biosciences Innsbruck (CMBI), Institute of Biochemistry, University of Innsbruck, Innrain 52, 6020 Innsbruck, Austria; 4Biopharma Austria, Process Science, Boehringer Ingelheim Regional Center Vienna GmbH & Co KG, Doktor-Boehringer-Gasse 5-11, 1121 Vienna, Austria; wolfgang.buchinger@boehringer-ingelheim.com

**Keywords:** bacteriophage-derived tag, fed-batch cultivation, proteolytic cleavage, inclusion body, caspase-2

## Abstract

Fusion protein technologies to facilitate soluble expression, detection, or subsequent affinity purification in *Escherichia coli* are widely used but may also be associated with negative consequences. Although commonly employed solubility tags have a positive influence on titers, their large molecular mass inherently results in stochiometric losses of product yield. Furthermore, the introduction of affinity tags, especially the polyhistidine tag, has been associated with undesirable changes in expression levels. Fusion tags are also known to influence the functionality of the protein of interest due to conformational changes. Therefore, particularly for biopharmaceutical applications, the removal of the fusion tag is a requirement to ensure the safety and efficacy of the therapeutic protein. The design of suitable fusion tags enabling the efficient manufacturing of the recombinant protein remains a challenge. Here, we evaluated several N-terminal fusion tag combinations and their influence on product titer and cell growth to find an ideal design for a generic fusion tag. For enhancing soluble expression, a negatively charged peptide tag derived from the T7 bacteriophage was combined with affinity tags and a caspase-2 cleavage site applicable for CASPase-based fusiON (CASPON) platform technology. The effects of each combinatorial tag element were investigated in an integrated manner using human fibroblast growth factor 2 as a model protein in fed-batch lab-scale bioreactor cultivations. To confirm the generic applicability for manufacturing, seven additional pharmaceutically relevant proteins were produced using the best performing tag of this study, named CASPON-tag, and tag removal was demonstrated.

## 1. Introduction

Currently, *Escherichia coli* is the production organism of choice for about 30% of all approved biopharmaceuticals, as it enables the fast and cost-efficient production of less complex biomolecules mainly due to high product titers in combination with rapid cell growth. Despite this, the expression of recombinant proteins with *E. coli* often results in the accumulation of target proteins into insoluble and/or nonfunctional aggregates known as inclusion bodies (IB) [1,2,3,4,5,6]. To overcome this bottleneck, a common strategy to improve soluble protein production is to express the protein of interest (POI) as a fusion protein [7,8]. Costa et al. provided a comprehensive overview of available and traditionally used fusion technologies [9]. Fusion partners change intrinsic properties such as solubility and stability or can even facilitate subsequent purification of the target protein [10,11]. However, a universally applicable fusion tag has yet to be discovered as the effectiveness of currently applied tags is highly dependent on the produced POI [8,12].

Apart from their benefits, fusion tags are also associated with negative consequences, for example, the activity of a POI can be affected by the interference of the fusion tag with other epitopes or domains of the recombinant protein [10,13]. Furthermore, it is a requirement for certain products such as biopharmaceuticals to remove the fusion partner during downstream processing, as the presence of the tag can induce immunogenicity in the patient [14]. The large molecular mass of many tags puts an additional burden on the cellular protein biosynthesis machinery, which reduces the expression performance of the target protein. In addition to that, large fusion tags also lead to an inherently decreased stoichiometric POI yield [10,15].

Over the last two decades, tagging proteins with N- or C-terminal affinity tags (e.g., His-tag, FLAG-tag, Strep-tag II [16,17,18]) has become increasingly popular to overcome the complexity of conventional purification by enabling affinity chromatography [19]. They provide a useful tool enabling easy purification and detection of POIs via the formation of strong bonds with chemical ligands. The most commonly used affinity tags are the Polyhistidine-tag (His-tag) and Strep-tag, which offer the advantages of having a small size and allow for fast and simple purification using immobilized metal affinity chromatography (IMAC) or Strep-Tactin chromatography, respectively [10,20]. However, a His-tag can have adverse effects on recombinant protein production, as the position and length of the tag can have a significant impact on the protein expression level [19,21]. It has also been reported that the use of the His-tag has negative effects on the solubility of the target protein [19].

Combining different fusion tags is an often-used strategy to balance these effects. For example, the incorporation of solubility-enhancing tags in tandem with an affinity tag yields a highly soluble POI that is easy to purify and detect [22]. In some cases, it has been reported that the addition of a fusion tag even improves overall expression levels [10,23,24]. A possible strategy to achieve an increase in overall recombinant protein expression is to fuse the sequence of a naturally highly expressed protein to the N-terminus of the POI [24]. However, this effect has also been observed with several other fusion tags including chaperone-like tags, acidic tags, and supercharged tags (e.g., polylysine tags) [13,23,25,26,27]. Negatively charged peptides from the T7 bacteriophage are highly attractive solubility tag candidates due to their small size and effectiveness [9,28]. It is hypothesized that the anionic charge of the nascent polypeptide leads to sufficient interprotein electrostatic repulsion, enabling the spontaneous folding of the protein into its native form and thereby preventing IB formation [28].

Another major challenge associated with biopharmaceuticals is the unwanted presence of the fusion partner linked with the target protein, as it may impede biological function or cause immunogenicity in the patient [10,14]. Therefore, reliable and efficient removal of the fusion partner is required. By inserting specific cleavage sequences recognized by proteases, the fusion tag can be removed during the purification process [16,29]. One such tag removal system is the recently developed CASPON technology, which is based on a modified human caspase-2 especially suited for the production of N-terminally tagged fusion proteins [30]. Its main advantages are the high activity, specificity, and good manufacturability of the enzyme, as well as the small size of the recognition sequence [30,31,32,33]. Great flexibility in the selection of suitable POIs is provided by the CASPON enzyme, due to the ability to cleave before any amino acid on the C-terminus of its recognition site [30,33]. This results in an authentic N-terminus, regardless of P1’ amino acid (notation according to Schechter and Berger [34]) when N-terminal fusion tags are used.

Human fibroblast growth factor-2 (hFGF-2) has several medical applications for regenerative therapy or tissue repair. It can be expressed in *E. coli* and several purification strategies have been published [35]. Addition of a hexa-histidine-tag (6-His-tag) to facilitate purification resulted in lower expression levels, which made this protein an ideal candidate for the improvement of fusion tag designs.

In this study, we designed different N-terminal combinatorial fusion tags by combining solubility- and affinity tags with caspase-2 cleavage sites. The chosen solubility tag is a short polyionic polypeptide that is based on the T7A3-tag, a sequence originating from the C terminus of the bacteriophage T7 gene 10B [28]. In order to obtain specific and efficient protease cleavage, the original T7A3-tag has been altered in two amino acid positions to alter a possible cleavage site for the use with caspase-2, resulting in the T7AC-tag. As affinity tags, 6-His- and StrepII-tag were selected due to their small size and high specificity. We evaluated the effect of different elements in these combinatorial fusion tags on hFGF-2 expression and fermentation characteristics. The generic applicability of the best performing tag tested in this study, named CASPON tag, was demonstrated with seven additional POIs.

## 2. Results and Discussion

### 2.1. Influence of Fusion Tag Elements in hFGF-2 Producing Fed-Batch Cultivations

To find a suitable design for a fusion tag that allows efficient manufacturing of POIs while considering both upstream and downstream processing aspects, we generated and evaluated several N-terminal fusion tag combinations (Figure 1). We investigated the influence of each fusion tag element on production strain growth and soluble expression of hFGF-2 in a series of identical carbon-limited fed-batch fermentations. The comparisons of soluble titers used here are solely based on untagged hFGF-2 concentrations taking the relative mass of the tag into account. For total titers (including fusion constructs) as well as stoichiometric POI content of all model proteins produced in this study, see Appendix A. To study all constructs in a comparative manner a reference fermentation was carried out using the untagged hFGF-2 construct 1 (Figure 1A). The resulting cell growth kinetics of this reference fermentation corresponded to the calculated values for the first 23 h of the fermentation (yellow trace in Figure 1B), after which the cells exhibited a reduced growth capability. This resulted in a reduction to 64 g cell dry mass (CDM)/L from the calculated value of 78 g CDM/L at the end of fermentation. The soluble recombinant hFGF-2 titer for the untagged construct reached 4.8 g/L at the end of fermentation. Interestingly, the fermentation producing native hFGF-2 was the only instance in the hFGF-2 experimental series, where the formation of IBs was observed. At the end of fermentation, the IB titer amounted to 2.1 g/L which is equivalent to 30% of the total protein production.

A fusion tag to facilitate subsequent downstream processing using an affinity tag and protease cleavage site was designed (construct 2 in Figure 1A). This tag consisted of a 6-His-tag coupled to a GSG linker and the caspase-2 cleavage site VDVAD. Construct 2 yielded a more than two-fold reduced recombinant hFGF-2 titer and an improved CDM concentration of 68 g/L at the end of fermentation. The soluble titer of construct 2 was 2.0 g/L, which corresponds to 1.9 g/L untagged hFGF-2 (Figure 1C).

To determine whether the origin of the negative effect on recombinant protein production of the above-mentioned fusion tag lies in the caspase-2 cleavage site, a construct containing solely the 6-His-tag and linker fused to hFGF-2 was tested (construct 3 in Figure 1A). Here, we observed that cellular growth was consistently lower than expected after induction (feed hour 14). Nonetheless, the CDM concentration at the end of the fermentation was nearly identical to the untagged hFGF-2 fermentation. However, a massive decrease in soluble recombinant protein titer by as much as 83%, down to 0.8 g/L, was observed compared to untagged hFGF-2. Therefore, the addition of the cleavage site in construct 2 resulted in a higher recombinant protein titer compared to construct 3. This suggests that the N-terminal 6-His-tag-linker combination alone has a highly negative impact on recombinant protein formation. While the tag used in construct 2 enables a simple downstream process, it appears to severely impede upstream processing. Decreases in protein solubility in His-tagged proteins have previously been reported [19]. As no IB formation was observed for the tagged hFGF-2 constructs used here, it can be concluded that the 6-His-tag solely affected the expression titer in this study. Since it has been shown that numerous solubility tags are not only capable of efficiently increasing the POI solubility but also enhancing the expression level of recombinant proteins [23], a solubility tag was tested in construct 4 to alleviate the negative impact seen in constructs 2 and 3 (Figure 1A). A short polyanionic solubility tag, originating from the minor capsid protein 10 B of the T7 bacteriophage was selected based on its size and net-charge properties [28]. Upon addition of this T7A3-tag, an increase in soluble hFGF-2 titer to 7.2 g/L was observed. This increase in volumetric titer is a result of higher cellular growth and specific titer. It constitutes a 50% increase relative to the untagged construct 1 and even surpasses the total hFGF-2 titer of construct 1 with the IB formation taken into account. However, the T7A3-tag harbors a potential cleavage site for caspase-2, which could interfere with downstream processing [30].

A modified tag, called T7AC-tag, with the mutations E17Q and E19Q to circumvent off-target cleavage by caspase-2 was used in construct 5. The T7AC-tag exhibited comparable expression-enhancing capabilities as the T7A3-tag since construct 5 yielded corresponding recombinant protein titer and cellular growth as observed with construct 4 (Figure 1C).

An additional StrepII-tag, enabling orthogonal affinity chromatography as well as additional off-line analytic procedures, was evaluated in construct 6. While cellular growth of this construct was comparable to constructs 4 and 5 (Figure 1A), the recombinant protein titer reached 6.2 g/L at the end of fermentation. This constitutes only a 30% increase relative to construct 1 (Figure 1A). However, one main contribution to this lower increase in pure recombinant protein titer compared to the T7AC and T7A3 tagged constructs can be attributed to the stoichiometric mass distribution between tag and POI, as this construct includes the largest tag design. Upon comparison of the total tagged protein titers at the end of fermentation, we see that the protein production is comparable with a total POI titer of 9.0 g/L for construct 4 (Figure 1A) and 8.2 g/L for the StrepII-tagged construct 6 (Figure 1A). In general, higher fusion tag mass leads to a lower relative yield of pure POI, and this effect is inherently more pronounced in small POIs such as hFGF-2.

Constructs 7 and 8 (Figure 1A) were used to evaluate the impact of the T7AC-tag with and without 6-His-tag. Construct 7, which harbors a 6-His-tag, reached soluble recombinant hFGF-2 titers of 9.0 g/L which corresponds to an increase of 88% compared to the untagged construct. Construct 8, further omitting the 6-His-tag, exhibited the highest soluble untagged hFGF-2 titer while exhibiting slightly improved cellular growth compared to all other constructs throughout the fermentation. The POI titers observed with this construct amounted to 10.5 g/L, corresponding to an increase of 120% compared to the reference cultivation. This illustrates that all fusion tag components besides the expression-enhancing T7 tags have a negative impact on the formation of recombinant hFGF-2. Even though construct 8 (Figure 1A) exhibits the most favorable upstream processing characteristics, the complete lack of any affinity tag or protease cleavage site severely limits its use.

The T7-polyanionic solubility tags used in this study have been shown to enhance the expression and solubility of hFGF-2. This is in accordance with previous literature, which demonstrated that short peptide tags can efficiently enhance the POI solubility, with the amino acids aspartic acid and glutamic acid being the main contributors [36]. The suggested mode of action of such small negatively charged peptide tags is to facilitate interprotein electrostatic repulsion which hinders the formation of IBs [28,36]. This causes an aggregation delay by which spontaneous folding of nascent proteins is promoted via chaperone-independent mechanisms [28].

In this study, we observed not only exclusively soluble production of hFGF-2 even at high recombinant protein titers, but also enhanced expression as a direct effect of the addition of the T7-tags. Similar behavior was reported with other solubility-enhancing tags such as the SUMO, NT11, MPB, and GST tags [25,27,37]. The mechanism by which these solubility tags enhance expression is currently not known with certainty. Hypotheses about the origin of this expression-enhancing characteristic include the possibility of enhanced transcript stability and modulation of the folding free energy of the mRNA translation initiation region (TIR) to a suitable level for ribosome-mRNA interactions [23,25]. Summarizing the fermentation results of the hFGF-2 constructs, we confirmed the previously reported negative impact of an N-terminal His-tag on recombinant protein production [21]. However, the inclusion of a bacteriophage-derived polyionic tag alleviated this unfavorable effect and lead to an increase in recombinant protein formation. Nonetheless, a negative effect can still be attributed to the 6-His-tag, even when coupled with the T7AC-tag, as construct 8 performed better in terms of pure POI formation than its counterpart construct 7 with the 6-His-tag. Construct 5 (Figure 1A), combining the T7AC-tag, 6-His-tag, and caspase-2 cleavage site features outstanding up- and downstream processing characteristics and was termed CASPON-tag. The volumetric POI yield was increased by 60% (Figure 1D) and the CASPON platform process can be utilized for purification. The fixed N-terminal region of the CASPON-tag allows for a tailor-made 5′ UTR independent from the POI, which warrants further investigation into the mechanism of its expression-enhancing effects.

### 2.2. Influence of Selected Fusion Tag Elements in Fed-Batch Cultivations of other POIs

To gain further insight into the expression enhancing capabilities of the CASPON-tag used in construct 5 (Figure 1A) and its general applicability for the CASPON platform process, its effects on other relevant biopharmaceutical proteins were studied. The CASPON-tag was tested on two additional model proteins (mature tumor necrosis factor-alpha, mTNF-α, and single-chain variable fragment BIWA4) and the performance of constructs with and without the CASPON-tag was evaluated (Figure 2).

Both constructs of each model protein (for a detailed description see Table 2) were expressed in identical fed-batch fermentations and their stoichiometric titer of the POI was compared. The non-T7-tagged version of mTNF-α (6-His_L_1__CS_mTNF-α) was expressed in the periplasm via ompA^SS^ leader as described by Fink et al. after expression in the cytoplasm did not yield detectable recombinant protein production [38]. For both model proteins, the total protein production increased upon the addition of the T7-tag. Especially for the non-T7-tagged mTNF-α, where no POI was detectable when expressed in the cytoplasm and 76% of the POI was produced in IBs in the periplasm, the solubilization and expression enhancing effect of the T7-tag can be clearly seen. In this case, the T7-tagged version of mTNF-α did not only enable the production in the cytoplasm but also proved to be highly effective at avoiding the formation of IBs.

In this study, BIWA4 was expressed using a process designed to generate IBs. The scFv is hereby expressed in the reducing environment of the cytoplasm, where the correct formation of the disulfide bonds of the scFv is hindered and IB formation is triggered [38]. As expected, due to the inability to form disulfide bonds, the POI aggregated as IB both with and without the T7-tag. Comparing the specific titers of the two BIWA4 constructs, the expression-enhancing effect of the T7-tag becomes obvious. This showcases that the expression-enhancing effect does not originate from the solubilization of IBs alone. The resulting expression enhancement of the T7-tagged POIs, mTNF-α, and BIWA was 1.4- and 2.3-fold, respectively.

The suitability of the cleavable T7-tagged construct 5 (Figure 1A), the CASPON-tag, was tested on five additional pharmaceutically relevant model proteins: granulocyte-colony stimulating factor (G-CSF), interferon γ (IFNγ), SARS-CoV-2 nucleocapsid protein (NP), parathyroid hormone (PTH), and recombinant human growth hormone (rhGH). Each of the recombinant proteins was expressed in a laboratory-scale glucose-limited fed-batch fermentation yielding excellent soluble recombinant protein titers. Soluble end titers of all fermentations, corrected for their fusion tags, as well as a comparison with reported titers of these biopharmaceuticals which are difficult to find in the literature for confidentiality reasons, are summarized in Table 1.

We showed that the CASPON-tag can be used to produce soluble G-CSF, IFNγ, NP, PTH, and rhGH. Excellent specific recombinant protein titers were obtained ranging from 45 mg/g CDM to 165 mg/g CDM for PTH and rhGH respectively, outperforming previously reported titers. Interestingly, no correlation between POI titers and molecular mass of the POI was observed (Appendix A). All POIs were produced by means of a standard fermentation, without previous fermentation optimization, thus implying that recombinant protein titers can be further enhanced by optimization of the fermentation process.

### 2.3. Tag Removal of CASPON-Tagged POIs

The applicability of the CASPON-tag for the soluble production of native pharmaceutically relevant proteins was shown by a cleavability study using a human caspase-2 variant [33]. Here, we demonstrated that all soluble-expressed POIs could be cleaved in an efficient and specific manner (Figure 3), indicating that the cleavage site is fully accessible by the protease for tag removal. The cleavage behavior of NP has already been reported by De Vos et al. and is therefore excluded here [44].

Even though the 6-His-tag is not located at the N-terminus of the CASPON-tag, accessibility is not affected, as shown by successful affinity purification and protease cleavage. Indeed, as shown in Figure 3, all POIs were successfully purified in one simple IMAC capture step. Figure 3 shows that very high cleavage yields are achievable with a circularly permuted human caspase-2 variant [33] for a variety of different POIs. Even though all of the presented POIs vary in their native N-terminal sequence (hFGF-2: Ala, mTNF-α: Val, rhGH: Phe, PTH: Ser, G-CSF: Ala, IFNγ: Gln), cleavability with caspase-2 was successfully demonstrated, yielding the native POIs, which is an excellent preposition for their biological activity. The faint band at ~9 kDa in the uncleaved PTH lane can be attributed to proteolytic degradation of the product during fermentation and subsequent co-purification. Another faint band observable in the cleaved IFNγ lane at ~12 kDa is due to both proteolytic degradation of the product during fermentation and minor off-target cleavage of the caspase-2 enzyme (data not shown). The data presented in this section demonstrates the general applicability of this enzyme for a generic platform process for recombinant protein production.

In conclusion, we have shown that the CASPON-tag is an essential part of the CASPON platform process. It facilitates high-titer production of POIs and simplifies downstream processing through its 6-His-tag and highly specific caspase cleavage site. This concludes that the CASPON-tag yields a highly capable system for manufacturing native pharmaceutical proteins in *E. coli*. Furthermore, it was demonstrated that the phage-derived T7AC solubility tag drastically enhances the expression of a host of model proteins in carbon-limited fed-batch fermentations.

## 3. Materials and Methods

### 3.1. Strains

All strains for cloning and expression purposes were purchased from New England Biolabs (NEB, Ipswich, MA, USA). Chemically competent *E. coli* NEB-5α cells were used for cloning purposes, while the *E. coli* strain BL21(DE3) was utilized for the expression of the recombinant proteins. Transformation and cultivation of the strains for cloning purposes were performed according to the manufacturer’s instructions.

### 3.2. Design of Expression Constructs

Enzymes and the Q5^®^ Site-Directed Mutagenesis Kit (SDMK) were purchased from NEB, all primers were acquired from Sigma Aldrich (St. Louis, MO, USA). For site-specific point mutations and deletions, the SDMK was used. The coding DNA of the substrate proteins hFGF-2, mTNF-α, BIWA4, rhGH, G-CSF, PTH, and IFNγ was codon-optimized with the GeneArt online tool (Thermo Fisher Scientific, Waltham, MA, USA). Expression constructs were cloned into an in-house produced pET30a*cer* expression vector under the control of a T7 promoter [45]. All cloning procedures were performed as described by Cserjan-Puschmann et al. [30]. In brief, the backbone was amplified using Q5^®^ High-Fidelity DNA Polymerase. Subsequently, the backbone and insert, which were purchased in gBlock format from Integrated DNA Technologies (IDT, Coralville, IA, USA), were digested with Ndel and Xhol restriction enzymes and ligated using T4 DNA ligase. All resulting plasmids were sequenced by Microsynth (Balgach, Switzerland) to confirm their correctness. The complete nucleotide and amino acid sequences of all expressed recombinant proteins can be found in Appendix A.

If not stated otherwise, the construction of the individual hFGF-2 constructs was performed as described above. Preparation of construct 2 was performed according to Lingg et al. [31]. Addition of the T7AC-tag (construct 5) was carried out according to Cserjan-Puschmann et al. [30]. Starting with these two constructs, small modifications such as deletions or additions of individual tag elements were introduced using site-directed mutagenesis. Primers 1 and 2 were used to mutate the T7AC-tag to the T7A3-tag. Primers 3 and 4 were employed to delete the caspase cleavage site, the linker L_1_, and the 6-His tag, yielding construct 8. The primers 5 and 6 erased the caspase cleavage site and the linker L_1_ resulting in construct 7. To obtain construct 2, primers 7 and 8 were used. The addition of the StrepII tag [20] and its accompanying linker (L_2_) with the amino acid sequence SA was performed via primers 9 and 10. The sequence of the StrepII tag was taken from Schmidt et al. [20]. If not stated otherwise, all additional model protein expression vectors were created as described above. The vector for SARS-CoV-2 nucleocapsid protein (NP) was assembled according to the workflow for pET30a*cer*-CASPON-NP as described by De Vos et al. [44]. The model proteins mTNF-α and rhGH were additionally fitted with an ompA signal sequence N-terminal of the CASPON tag for periplasmatic expression. All constructs tested in addition to hFGF-2 are depicted in Table 2.

### 3.3. Lab-Scale Fed-Batch Cultivations

All procedures and materials relevant to the fermentation processes including media preparation and its composition are described by Cserjan-Puschmann et al. [30]. A glucose yield coefficient Y_X/S_ of 0.33 g/g was employed based on the findings by Marisch et al. [46]. For each model protein, one fermentation process was chosen and repeated in an identical fashion for all of its tag variations. All model protein fermentations were performed once and reproducibility of the carbon-limited fed-batch fermentation was tested with construct 8 which was performed in duplicate (Appendix A). hFGF-2 was expressed using a glucose-limited fed-batch fermentation process as described by Lingg et al. [31]. Briefly, cells were grown in a 20 L (4 L batch volume, 8 L feed volume) computer-controlled (Siemens PS7, Siemens WinCC, Siemens AG, Munich, Germany) bioreactor (Bioengineering, Wald, Switzerland) equipped with standard probes (temperature, pH, dissolved oxygen (pDO)). The pH value was set to 7.0 ± 0.05 while temperature was held at 37 ± 0.05 °C throughout the batch phase and was shifted to 30 ± 0.05 °C at the start of the feed phase. To ensure aerobic fermentation conditions, the pDO value was maintained at >30% (*v*/*v*) throughout the process. Induction was carried out by the addition of 0.9 µmol Isopropyl-β-D-thiogalactopyranoside per gram of calculated fermentation-end CDM after two generations of feed. mTNF-α and BIWA4 were produced analogously in an identical bioreactor with 30 L working volume according to the parameters described in the Appendix A. Production of rhGH, G-CSF, PTH, NP, and IFN-γ was carried out using a glucose-limited fed-batch process in a benchtop fermentation device according to Fink et al. [38]. NP was produced according to De Vos et al. [44]. All fermentation parameters not described elsewhere are listed in the Appendix A.

### 3.4. Product Analysis

Cell disruption for intracellular product analysis, as well as separation of soluble and insoluble POI fractions, were carried out according to Fink et al. with NuPAGE Sample Reducing Agent (10x) (Invitrogen, Waltham, MA, USA) added additionally to the lysis buffer at a concentration of 4 mmol L^−1^ [47]. Estimation of recombinant protein titre was performed via reducing SDS-polyacrylamide electrophoresis (PAGE) following the protocols provided by Stargardt et al. and Cserjan-Puschmann et al. [30,45]. For semi-quantitative determination of POI titers, 15 µL of sample mix consisting of 65% sample or standard, 25% NuPAGE LDS Sample Buffer (4x), and 10% NuPAGE Reducing Agent (10x) were heated to 70 °C for 10 min. After which the samples were loaded onto a PAGE gel (Invitrogen NuPAGE 4–12% Bis-Tris) and the proteins were separated by applying a constant voltage of 200 V for 45 min. The gels were stained with Coomassie R250 and the POI concentration was estimated by densitometry analysis using ImageQuantTL 1D (Version 7.0). The same acceptance criteria outlined by Lingg et. al. [31] were applied. As quantification standards, Bovine Serum Albumin heat shock fraction (Sigma Aldrich, St. Louis, MO, USA) was used at concentrations of 75, 50, and 25 µg/mL.

### 3.5. Downstream Processing and Removal of Fusion Tags

After fermentation, cells were harvested by centrifugation and digested by high-pressure homogenization. Chromatographic purification was performed after clarification of the homogenate via centrifugation and subsequent filtration. All proteins were captured and buffer exchanged into PBS as described by Lingg et al. [31]. The purified fusion proteins were then cleaved for 2 h (hFGF-2, mTNF-α, rhGH), 3 h (PTH), or 24 h (G-CSF, IFNγ) with a variant of circularly permutated caspase-2 [30] in a molar enzyme to substrate ratio of 1:100 (hFGF-2, mTNF-α, HGH, PTH, G-CSF) or 1:10 (IFN). All buffers and cleavage conditions are described elsewhere [31]. The resulting samples were analyzed via reducing 1D SDS-PAGE as described above.

## Figures and Tables

**Figure 1 ijms-23-07678-f001:**
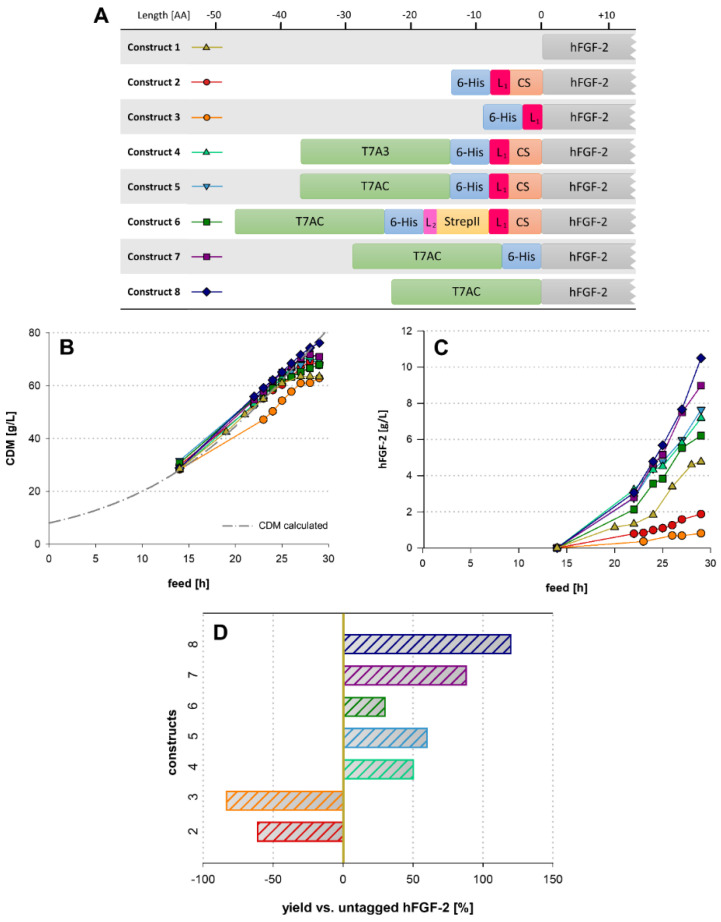
Panel (**A**): Schematic representation of all hFGF-2 constructs and tag variations; L stands for linker while CS denotes the caspase-2 cleavage site; the number of amino acids in the tag constructs is displayed by the ruler above; complete sequences of all tag elements are listed in Appendix A. Panel (**B**): cell growth in the course of time. Panel (**C**): Volumetric soluble protein titer throughout the fermentation process of the tag constructs. Panel (**D**): Ratio of stoichiometric POI yields at the end of the fermentations. The 0% line represents the titer of untagged hFGF-2. All values represent stoichiometric POI yields.

**Figure 2 ijms-23-07678-f002:**
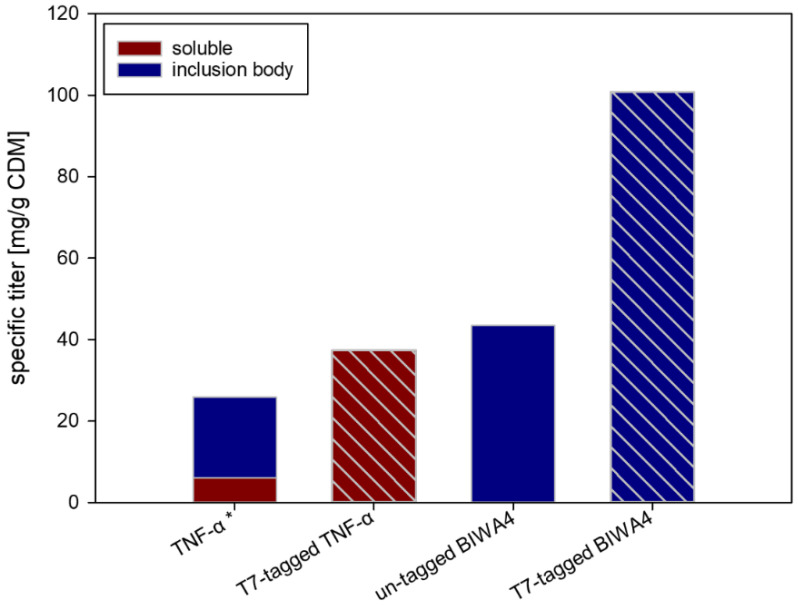
Comparison of specific fermentation end titers of TNF-α and BIWA4. Textured bars represent the T7-tagged constructs. Fusion protein sequences are as follows: 6His_L_1__CS_TNF-α (TNF-α), CASPON_TNF-α (T7-tagged TNF-α), CASPON_BIWA4 (T7-tagged BIWA4). The star (*) denotes constructs that have been expressed in the periplasm with ompA^SS^. Constructs are described in detail in Table 2. All values represent stoichiometric pure POI yields.

**Figure 3 ijms-23-07678-f003:**
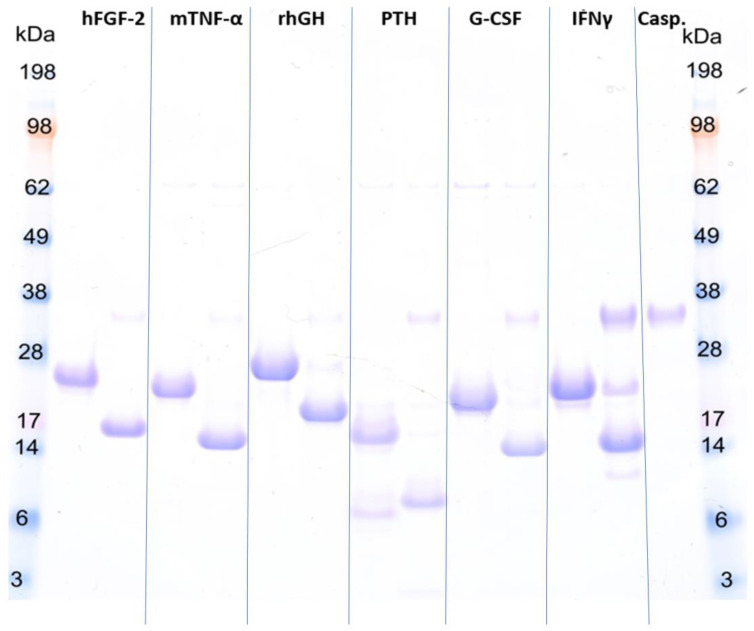
SDS Page of the enzymatic cleavage of produced fusion proteins. Protein identities are denoted at the top of the gel. For each model protein, two lanes are reserved. In the first lane, the uncleaved protein is shown while the cleaved protein can be seen in the second lane. In the lane titled Casp. the pure caspase-2 used for proteolytic cleavage is shown. Cleavage conditions were optimized for each protein. The molar ratio of enzyme to substrate was either 1:100 (hFGF-2, mTNF-α, rhGH, PTH, G-CSF) or 1:10 (INFγ). The cleavage reaction was terminated when either full substrate cleavage was achieved or a reaction time of 24 h was reached. Detailed cleavage conditions are given in Section 3.5.

**Table 1 ijms-23-07678-t001:** Expression of POIs with and without CASPON-tag. Values for untagged POIs are taken from the literature. Soluble end titres of tested fusion proteins are adjusted for the untagged POIs. The rows titled Untagged POI depict specific or volumetric *E. coli* fermentation titres reported in the literature. Values denoted with a star (*) originate from shaker flask cultures and were in case of PTH corrected for the untagged protein content as it was expressed by Fu et al. as Trx-hPTH fusion protein [39].

	G-CSF	IFNγ	NP	PTH	rhGH
***Tagged POI* (mg/g CDM)**	67	46	106	45	165
***Untagged POI* (mg/g CDM)**					63 [40]
***Tagged POI* (g/L)**	1.9	1.5	3.6	1.5	4.4
***Untagged POI* (g/L)**	0.16 * [41]	0.07 * [42]	0.05 * [43]	0.35 * [39]	

**Table 2 ijms-23-07678-t002:** Additionally tested model proteins with corresponding expression constructs.

Model Protein	Expression Construct
*Mature Tumor Necrosis Factor α*	pET30a*cer*_ompA^SS^_6-His_L_1__CS_mTNF-α
*T7-tagged Mature Tumor Necrosis Factor α*	pET30a*cer*_CASPON_mTNF-α
*BIWA4*	pET30a*cer*_BIWA4
*T7-tagged BIWA4*	pET30a*cer*_CASPON_BIWA4
*Recombinant Human Growth Hormone*	pET30a*cer*_ompA^SS^_CASPON_rhGH
*Granulocyte Colony Stimulating Factor*	pET30a*cer*_CASPON_G-CSF
*Parathyroid Hormone*	pET30a*cer*_CASPON_PTH
*SARS-CoV-2 Nucleocapsid Protein*	pET30a*cer*_CASPON_NP
*Interferon γ*	pET30a*cer*_CASPON_IFN-γ

## Data Availability

Not applicable.

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
