# Peer review of "Fusion Tag Design Influences Soluble Recombinant Protein Production in Escherichia coli"

_ijms, 2022, doi:10.3390/ijms23147678_

Round 1
Reviewer 1 Report
Cserjan-Puschmann et al fused various peptide tags to proteins of interest in order to improve their recombinant expression and purification from E. coli cells. Although the efforts are appreciated, the limited data in this work are not sufficient to support their bold conclusion that the CASPON-tag is the best performing tag.
1. From the description of the experiments and results, it is not clear if the experiments were ever repeated. No error bars nor statistical analyses were performed.
2. It is unclear to this reviewer how exactly the protein yields were quantified, given the super brief methods authors outlined. It is thus impossible to judge if the results can support their conclusions.
3. Authors only compared the CASPON tag with very limited amount of other tags, how can authors make the bold conclusion that this CASPON tag is the best?
4. Authors only fused the tag at the N term of proteins, while there are many examples that tagging proteins at this terminus would cause problems. So what happens if the tags were fused to the C term?
5. Authors showed gels as an evidence of tagged/cleaved proteins purified, but do we know if those proteins are biologically functional? This is critical as authors claimed that these would be of pharmaceutical interests.
6. One would like to see the comparisons of the different tags when using standard small scale cell cultures in the lab, not the fermentation approach which is hard to replicate by others.
Author Response
Dear Reviewer,
thank you for your careful review and suggestions on how to enhance the clarity and significance of our manuscript. We have considered and answered all of your comments to our best knowledge and made some adaptions to the manuscript according to your comments. In the following we would like to respond to your input point by point.
Point 1: From the description of the experiments and results, it is not clear if the experiments were ever repeated. No error bars nor statistical analyses were performed.
Response 1: We thank the reviewer for the comment and acknowledge that this information is missing from the manuscript. We therefore added the statement in line 367 to read “All model protein fermentations were performed once and reproducibility of the carbon limited fed-batch fermentations was tested with construct 8 which was performed in duplicate (supplementary materials Figure S1).” We agree that the reproducibility must be shown and added the information in the supplementary materials Figure S1 (page 11). Unfortunately, due to the complex and laborious nature of the controlled lab-scale fermentations, it was not possible to perform further repetitions of the fermentations. However, because all relevant process parameters during the fermentations are constantly monitored and computer controlled, we can be reasonably certain that the experiments are highly reproducible (as is shown in Figure S1).
Point 2: It is unclear to this reviewer how exactly the protein yields were quantified, given the super brief methods authors outlined. It is thus impossible to judge if the results can support their conclusions.
Response 2: We have attempted to avoid repetition and keep the methods description short, but the authors agree that the description of the protein quantification methodology is too brief, and more insight is needed. Therefore, starting in line 393 the following section was added: “For semi-quantitative determination of POI titers, 15 µL of sample mix consisting of 65% sample or standard, 25% NuPAGE LDS Sample Buffer (4x) and 10% NuPAGE Reducing Agent (10x) were heated to 70 °C for 10 min. After which the samples were loaded onto a PAGE gel (Invitrogen NuPAGE 4-12% Bis-Tris) and the proteins sepa-rated by applying a constant voltage of 200 V for 45 min. The gels were stained with Coomassie R250 and the POI concentration estimated by densitometry analysis using ImageQuantTL 1D (Version 7.0). The same acceptance criteria outlined by Lingg et. al. [30] were applied.”
Point 3: Authors only compared the CASPON tag with very limited amount of other tags, how can authors make the bold conclusion that this CASPON tag is the best?
Response 3: We agree with the reviewer that the conclusion that the CASPON tag is the best fusion tag is formulated too boldly. Therefore, the following lines have been adapted to specify that the CASPON tag is the best performing tag tested in this study.
Line 30: “To confirm the generic applicability for manufacturing, seven additional pharmaceutically relevant proteins were produced using the best performing tag of this study, named CASPON-tag, and tag removal was demonstrated.”
Line 109: “The generic applicability of the best performing tag tested in this study, named CASPON tag, was demonstrated with seven additional POIs.” The authors like to state that the limited number of tested tags is due to the high standards of our approach as we wanted to create a fusion tag with minimal stoichiometric impact on recombinant protein yield and most other fusion tags do not fulfill this requirement. Therefore, we limited our screening for additional fusion tags as we already found an already excellent performing fusion tag with the T7AC tag.
Point 4: Authors only fused the tag at the N term of proteins, while there are many examples that tagging proteins at this terminus would cause problems. So what happens if the tags were fused to the C term?
Response 4:
We agree with the reviewer that investigations of C terminal fusion of our tested tag variants would be of interest. However, our CASPON technology is limited to N-terminal fusion proteins, as the human Caspase-2 cleaves on the C terminus of its recognition sequence. Therefore, to obtain the native recombinant protein, all our fusion tag constructs need to be fused to the N-terminus of the protein of interest.
To enhance the clarity of the manuscript regarding this point the following sentences were adapted:
Line 88: “One such tag removal system is the recently developed CASPON technology, which is based on a modified human caspase-2 especially suited for the production of N-terminally tagged fusion proteins [1].”
Line 92: “Great flexibility in the selection of suitable POIs is provided by the CASPON enzyme, due to the ability to cleave before any amino acid on the C-terminus of its recognition site [1,2].“
Line 94: “This results in an authentic N-terminus, regardless of P1' amino acid (notation according to Schechter and Berger [34]) when N-terminal fusion tags are used.”
Furthermore, Zhang et. al. [3] showed that the T7A3 tag enhances the solubility of recombinant proteins when fused to the C-terminus. However, they did not report an increase in recombinant protein titer. Therefore, the authors infer that while C-terminal fusions of the T7A3 tag lead to enhanced solubility of the POI, the expression enhancement observed in this work is a characteristic of N-terminal fusions.
Point 5: Authors showed gels as an evidence of tagged/cleaved proteins purified, but do we know if those proteins are biologically functional? This is critical as authors claimed that these would be of pharmaceutical interests.
Response 5: The authors acknowledge that the biological activity of the produced proteins is an important remark and must be tested. However, the project, wherein this research is embedded is still ongoing and activity assays are scope of the project in the future, the data of which we hope to publish separately. For now, we cannot comment on the activity of the produced proteins, but we have shown the cleavability of our N-terminal fusion proteins, which yield native N-termini and therefore providing the best pre-requisite for biological activity. The following statement has been changed to clarify this:
Line 301 “Even though all of the presented POIs vary in their native N-terminal sequence (hFGF-2: Ala, mTNF-α: Val, rhGH: Phe, PTH: Ser, G-CSF: Ala, IFNγ: Gln), cleavability with caspase-2 was successfully demonstrated, yielding the native POIs, which is an excellent preposition for their biological activity.”
Point 6: One would like to see the comparisons of the different tags when using standard small scale cell cultures in the lab, not the fermentation approach which is hard to replicate by others.
Response 6: We agree with the reviewer, that standard small scale cell cultures are more easily reproducible by others. However, since we aim to produce fusion proteins for their ultimate use as biopharmaceuticals, we believe that the fermentation approach we used is far more representative of the processes used in the pharmaceutical industry. This approach was chosen together with our industrial partner, as it facilitates ease of process upscaling while still delivering representative data for production scale, which is often not the case with small scale cell cultures. Moreover, our fermentations are computer controlled, where critical parameters such as oxygen concentration, pH, temperature and nutrient feed are constantly monitored and adjusted. Whereas in small scale cell culture all these parameters are not controlled due to its inherent limitations. The authors therefore conclude that while small scale cell culture is more easily replicable by others, our fermentation approach yields a more robust and reproducible process.
- Cserjan-Puschmann, M.; Lingg, N.; Engele, P.; Kröß, C.; Loibl, J.; Fischer, A.; Bacher, F.; Frank, A.-C.; Öhlknecht, C.; Brocard, C.; et al. Production of Circularly Permuted Caspase-2 for Affinity Fusion-Tag Removal: Cloning, Expression in Escherichia coli, Purification, and Characterization. Biomolecules 2020, 10, doi:10.3390/biom10121592.
- Lingg, N.; Kröß, C.; Engele, P.; Öhlknecht, C.; Köppl, C.; Fischer, A.; Lier, B.; Loibl, J.; Sprenger, B.; Liu, J.; et al. CASPON platform technology: Improved circularly permuted caspase-2 cleaves tagged fusion proteins before all 20 natural amino acids at the N-terminus , Manuscript submitted for publication.
- Zhang, Y.B.; Howitt J Fau - McCorkle, S.; McCorkle S Fau - Lawrence, P.; Lawrence P Fau - Springer, K.; Springer K Fau - Freimuth, P.; Freimuth, P. Protein aggregation during overexpression limited by peptide extensions with large net negative charge. Protein Expr Purif. 2004, 36, 207-216.
Reviewer 2 Report
Fusion tags are routinely used in expression of recombinant proteins expression in prokaryotes and some eukaryotes systems to improve expression, detection, purification and activity. This publication by Köppl et al. provides insight into impact of the recently developed CASPON system on recombinant protein production. This manuscript is interesting but can be improved by some minor changes.
A) One missing control is the StepII only construct without T7AC like construct 2 or 3 to understand the impact on other fusion tags on the protein expression? Authors have clearly identified His tag doesn't work for this protein, but other tags may work.
B) In figure 1D, can authors label the Y-axis with the construct numbers to make it easier for readers to understand the results.
B) In figure 3, can the authors comment on what the faint bands on uncleaved PTH and cleaved IFNy lanes.
C) The authors tested many proteins for recombinant proteins, can the authors comment on the different sizes of proteins tested? Did the authors observe the impact of protein size resulting in reduced protein expression?
C) Spelling mistake on line 291: achieved.
D) Formatting errors on lines 217 and 256.
Author Response
Dear Reviewer,
thank you for your careful review and suggestions on how to enhance the clarity and significance of our manuscript. We have considered and answered all of your comments to our best knowledge and made some adaptions to the manuscript according to your comments. In the following we would like to respond to your input point by point.
Fusion tags are routinely used in expression of recombinant proteins expression in prokaryotes and some eukaryotes systems to improve expression, detection, purification and activity. This publication by Köppl et al. provides insight into impact of the recently developed CASPON system on recombinant protein production. This manuscript is interesting but can be improved by some minor changes.
Point 1: One missing control is the StepII only construct without T7AC like construct 2 or 3 to understand the impact on other fusion tags on the protein expression? Authors have clearly identified His tag doesn't work for this protein, but other tags may work.
Response 1: The authors agree that these constructs would be of interest to test as possible alternatives for the His tag. However, no negative impact apart from the larger mass of the fusion tag and therefore lower corrected recombinant POI titer was observed in construct 6 compared to construct 5 and 4. Another reason we omitted these experiments was the nature of the project itself regarding the DSP purification, which is already established using the His-tag chromatography resin which is more easily available and scale-up is more economical. Therefore, using the StrepII tag alone as affinity tag would not have been an option in the constraints of this project.
Point 2: In figure 1D, can authors label the Y-axis with the construct numbers to make it easier for readers to understand the results.
Response 2: We thank the reviewer for this remark and agree that the quality of the figure would benefit from the change. Therefore, the suggestion was added to figure 1D.
Point 3: In figure 3, can the authors comment on what the faint bands on uncleaved PTH and cleaved IFNy lanes.
Response 3: The authors acknowledge that the clarity of the manuscript would increase upon addition of further information on the faint band observable in figure 3. Therefore, the following lines have been added to the figure description: Line 305 “The faint band at ~9 kDa in the uncleaved PTH lane can be attributed to proteolytic degradation of the product during fermentation and subsequent co-purification. Another faint band observable in the cleaved IFNγ lane at ~12 kDa is due to both proteolytic degradation of the product during fermentation and minor off-target cleavage of the caspase-2 enzyme (data not shown).”
Point 4: The authors tested many proteins for recombinant proteins, can the authors comment on the different sizes of proteins tested? Did the authors observe the impact of protein size resulting in reduced protein expression?
Response 4: We agree with the reviewer, that the size of the recombinant proteins could have an influence on the recombinant protein expression. However, in this study we have not seen any direct correlation between recombinant protein titers and POI size. The authors have evaluated proteins ranging in size from 13.7 kDa for CASPON-PTH up to 49.8 kDa for CASPON-NP. The different protein masses of all tested variants as well as their titers are given in the supplementary materials table S3.
The paragraph starting at line 278 was adapted to read: “Interestingly, no correlation between POI titers and molecular mass of the POI was observed (supplementary materials table S3). All POIs were produced by means of a standard fermentation, without previous fermentation optimization, thus implying that recombinant protein titers can be further enhanced by optimization of the fermentation process.”
Point 5: Spelling mistake on line 291: achieved.
Response 5: The authors thank the reviewer for the thorough review and have corrected the mistake.
Point 6: Formatting errors on lines 217 and 256.
Response 6: The authors thank the reviewer for the thorough review and have corrected the formatting errors.
Reviewer 3 Report
ijms-1778311-peer-review-v1 Comments
Authors investigated N-terminal fusion tag combinations and their effect on product titer and cell growth to find an ideal design for a generic protein fusion that enhances solubility, without any negative consequences to cell fitness. For enhancing soluble expression, a negatively charged peptide tag derived from the T7 bacteriophage was combined with affinity tags and a caspase-2 cleavage site to enable separation of the tag from the target protein. Examination of the best performing tag is discussed.
The work merits publication upon revision in light of the comments below.
Major Comment
(1) First sentence of the abstract “Fusion protein technologies to facilitate soluble expression, detection, or subsequent affinity purification in Escherichia coli are widely used but are often associated with negative consequences” brought to mind the recent work of Betterle et al. (2020 Cyanobacterial production of biopharmaceutical and biotherapeutic proteins. Front. Plant Sci. 11:237. https://doi.org/10.3389/fpls.2020.00237), where they demonstrated fusion protein technologies that facilitate soluble over-expression, detection, and subsequent affinity purification in the cyanobacterium Synechocystis, without any negative consequences to cell fitness. Authors ought to discuss this relevant publication in their paper.
(2) It would be of general interest to readers to know the amount of glucose, or total organic carbon consumed for the generation of the CDW, e.g., by supplementing the results of Figure 1B with these additional results. This information would help to complete the stoichiometries discussed in this work.
Minor Comments:
(a) Materials and methods subsection title “3.2 Creation of expression constructs” should be revised to “3.2 Design of expression constructs.”
(b) Thank you for reporting titer results as weight (g) product per weight (g) CDW, rather than weight (g) product per L culture volume.
Author Response
Dear Reviewer,
thank you for your careful review and suggestions on how to enhance the clarity and significance of our manuscript. We have considered and answered all of your comments to our best knowledge and made some adaptions to the manuscript according to your comments. In the following we would like to respond to your input point by point.
Major Comment
Point 1: First sentence of the abstract “Fusion protein technologies to facilitate soluble expression, detection, or subsequent affinity purification in Escherichia coli are widely used but are often associated with negative consequences” brought to mind the recent work of Betterle et al. (2020 Cyanobacterial production of biopharmaceutical and biotherapeutic proteins. Front. Plant Sci. 11:237. https://doi.org/10.3389/fpls.2020.00237), where they demonstrated fusion protein technologies that facilitate soluble over-expression, detection, and subsequent affinity purification in the cyanobacterium Synechocystis, without any negative consequences to cell fitness. Authors ought to discuss this relevant publication in their paper.
Response 1: The authors agree that the listed publication is of relevance to the field and regret that they have not referenced it. However, Betterle et al. described that the cpc operon, where they inserted the gene of interest, can be disrupted or even deleted without adverse effects on cellular fitness, which is also backed by literature, but they did not describe consequences due to the expression of fusion proteins on cellular fitness. They give no information on the cellular growth of their cultures carrying the fusion constructs other than that the cells were harvested at an OD730 of 1.We agree that the applied technique of using a highly expressed host cell protein as N-terminal fusion partner is shown by Betterle et al. as very effective and is ought to be discussed in our manuscript.
Therefore, the paragraph starting at line 72 has been adapted to read: “In some cases, it has been reported that the addition of a fusion tag even improves overall expression levels [10,23,24]. A possible strategy to achieve an increase in overall recombinant protein expression is to fuse the sequence of a naturally highly expressed protein to the N-terminus of the POI [24]. However, this effect has also been observed with several other fusion tags including chaperone-like tags, acidic tags, and supercharged tags (e.g.: polylysine tags) [13,23,25-27].”
Additionally, the statement starting in line 15 has ben altered to read “Fusion protein technologies to facilitate soluble expression, detection, or subsequent affinity purification in Escherichia coli are widely used but may also be associated with negative consequences.”
Point 2: It would be of general interest to readers to know the amount of glucose, or total organic carbon consumed for the generation of the CDW, e.g., by supplementing the results of Figure 1B with these additional results. This information would help to complete the stoichiometries discussed in this work.
Response 2: We agree that the amount of consumed glucose during the fermentation is of interest for the readers. In this work a glucose yield coefficient of 0.33 grams CDM per gram of glucose was assumed for media preparation and calculation of the theoretical CDM. This yield coefficient is taken from Marisch et al. [1] who determined it to be between 0.34 and 0.4 grams per gram of glucose for an uninduced E. coli culture. Since most of our cultures followed the calculated growth curves quite well given the additional burden recombinant protein production exerts, the residual glucose in the fermentation broth was not determined and considered negligible.To give additional clarity to the manuscript, line 364 was added to read: “A glucose yield coefficient YX/S of 0.33 g/g was employed based on the findings by Marisch et al. [46].”
Minor Comments:
Point a: Materials and methods subsection title “3.2 Creation of expression constructs” should be revised to “3.2 Design of expression constructs.”
The authors acknowledge that “Design of expression constructs” is a more fitting title for the subsection and changed the heading in line 327.
Point b: Thank you for reporting titer results as weight (g) product per weight (g) CDW, rather than weight (g) product per L culture volume.
We thank reviewer 3 for this comment.
- Marisch, K.; Bayer, K.; Cserjan-Puschmann, M.; Luchner, M.; Striedner, G. Evaluation of three industrial Escherichia coli strains in fed-batch cultivations during high-level SOD protein production. Microbial Cell Factories 2013, 12, 58, doi:10.1186/1475-2859-12-58.
Reviewer 4 Report
Herein authors present a paper describing a combinatorial approach for proper production of heterologous protein in E. coli.
The manuscript is nice, i only suggest, if possible to check the suitability of this system as C-terminal fusion tag.
In addition a few typos are present as in line 256
Author Response
Dear Reviewer,
thank you for your careful review and suggestions on how to enhance the clarity and significance of our manuscript. We have considered and answered all of your comments to our best knowledge and made some adaptions to the manuscript according to your comments. In the following we would like to respond to your input point by point.
Point 1: The manuscript is nice, i only suggest, if possible to check the suitability of this system as C-terminal fusion tag.
Response 1: We agree with the reviewer that investigations of C terminal fusion of our tested tag variants would be of interest. However, our CASPON technology is limited to N-terminal fusion proteins, as the human Caspase-2 cleaves on the C terminus of its recognition sequence. Therefore, to obtain the native recombinant protein, all our fusion tag constructs need to be fused to the N-terminus of the protein of interest.
To enhance the clarity of the manuscript regarding this point the following sentences were adapted:
Line 88: “One such tag removal system is the recently developed CASPON technology, which is based on a modified human caspase-2 especially suited for the production of N-terminally tagged fusion proteins [1].”
Line 92: “Great flexibility in the selection of suitable POIs is provided by the CASPON enzyme, due to the ability to cleave before any amino acid on the C-terminus of its recognition site [1,2].“
Line 94: “This results in an authentic N-terminus, regardless of P1' amino acid (notation according to Schechter and Berger [34]) when N-terminal fusion tags are used.”
Point 2: In addition, a few typos are present as in line 256
Response 2: The authors thank the reviewer for the comment and have corrected the errors.
- Cserjan-Puschmann, M.; Lingg, N.; Engele, P.; Kröß, C.; Loibl, J.; Fischer, A.; Bacher, F.; Frank, A.-C.; Öhlknecht, C.; Brocard, C.; et al. Production of Circularly Permuted Caspase-2 for Affinity Fusion-Tag Removal: Cloning, Expression in Escherichia coli, Purification, and Characterization. Biomolecules 2020, 10, doi:10.3390/biom10121592.
- Lingg, N.; Kröß, C.; Engele, P.; Öhlknecht, C.; Köppl, C.; Fischer, A.; Lier, B.; Loibl, J.; Sprenger, B.; Liu, J.; et al. CASPON platform technology: Improved circularly permuted caspase-2 cleaves tagged fusion proteins before all 20 natural amino acids at the N-terminus , Manuscript submitted for publication.
- Zhang, Y.B.; Howitt J Fau - McCorkle, S.; McCorkle S Fau - Lawrence, P.; Lawrence P Fau - Springer, K.; Springer K Fau - Freimuth, P.; Freimuth, P. Protein aggregation during overexpression limited by peptide extensions with large net negative charge. Protein Expr Purif. 2004, 36, 207-216.

Round 2
Reviewer 1 Report
Authors have adequately addressed my concerns.
Reviewer 3 Report
Authors revised their paper in accordance with the reviewer comments.